# Hepatic and Pulmonary Vasoactive Response Triggered by Potentially Hazardous Chemicals After Passing Through the Gut Mucosa

**DOI:** 10.3390/diagnostics15192444

**Published:** 2025-09-25

**Authors:** Mircea Dragoteanu, Ștefan Tolea, Ioana Duca, Raluca Mititelu, Kalevi Kairemo

**Affiliations:** 1Department of Nuclear Medicine, “Professor Dr. Octavian Fodor” Regional Institute for Gastroenterology and Hepatology, 400162 Cluj-Napoca, Romania; 2College of Pharmacy, Ferris State University, Big Rapids, MI 49307, USA; stefan.tolea@gmail.com; 3Department of Internal Medicine, REHA Zentrum, 5020 Salzburg, Austria; ioanaducagrigorescu@gmail.com; 4Nuclear Medicine Department, University of Medicine and Pharmacy “Carol Davila”, 050474 Bucharest, Romania; ralunuclear@yahoo.com; 5Central Military Hospital “Carol Davila”, 010825 Bucharest, Romania; 6Department of Theranostics, Docrates Cancer Center, 00180 Helsinki, Finland; kalevi.kairemo@gmail.com; 7Department of Nuclear Medicine, The University of Texas MD Anderson Cancer, Houston, TX 77030, USA

**Keywords:** vasoconstriction, portal flow, pulmonary blood flow, per-rectal portal scintigraphy, potentially hazardous chemicals, defensive response, colorectal cancer, early metastasis

## Abstract

**Background/Objectives**: In a previous study, we observed significantly prolonged hepatic and pulmonary first-pass transit times (TTs) for ^99m^Tc-pertechnetate absorbed through the colorectal mucosa during per-rectal portal scintigraphy (PRPS). This decrease in radiotracer flow velocity was not seen when ^99m^Tc-pertechnetate was administered into the spleen during trans-splenic portal scintigraphy or injected intravenously in radionuclide angiocardiography. We hypothesized that ^99m^Tc-pertechnetate, an artificial compound, is recognized during colorectal absorption as a potentially hazardous chemical (PHC), with its hepatic and pulmonary slowdown aiding elimination. A similar sudden decrease in portal flow occurs during early metastasis of colorectal cancer (CRC), as shown by a pathological rise in the hepatic perfusion index. We aimed to study the hepatic and pulmonary vasoactive responses triggered by PHCs after they pass through the gut mucosa and evaluate the potential activation of this mechanism in early CRC metastasis. **Methods**: We measured transit times to determine whether hepatic and pulmonary vasoconstriction occur in response to radiotracers administered at different sites. We performed PRPS with in vivo ^99m^Tc-labelled RBC to evaluate the liver transit time (LTT) and right heart to liver circulation time (RHLT). Liver angioscintigraphy (LAS) was used to assess RHLT following the intravenous injection of ^99m^Tc-pertechnetate and ^99m^Tc-HDP (hydroxyethylene-diphosphate). Lower rectum transmucosal dynamic scintigraphy (LR-TMDS) was conducted to measure RHLT of ^99m^Tc-pertechnetate delivered into the lower rectum submucosa. LAS was performed to assess LTT for ^99m^Tc-HDP intravenously injected and delivered to the gut mucosa via arterial flow. **Results:** In healthy volunteers, PRPS showed notably increased LTT, ranging from 23.5 to 25.5 s, and RHLT (between 39.5 and 42.5 s) for in vivo ^99m^Tc-labelled RBC. Significantly lower RHLT values ranging from 9 to 13.5 were observed for ^99m^Tc-pertechnetate and ^99m^Tc-HDP administered intravenously during LAS, as well as for ^99m^Tc-pertechnetate at LR–TMDS (between 12 and 15 s). The LTT assessed at LAS for ^99m^Tc-HDP ranged from 22 to 27 s. **Conclusions**: An intense vasoconstriction occurs in the liver and lungs in response to substances recognized by the body as PHCs when they pass through the gut mucosa, aiding their elimination.

## 1. Introduction

^99m^Tc-pertechnetate absorbed from the upper rectum and terminal colon in per-rectal portal scintigraphy (PRPS) shows a three- to fourfold increase in the liver transit time (LTT) from the portal vein to the right atrium. A similar increase was observed at PRPS in the circulation time from the right heart to the liver (RHLT), mainly involving pulmonary transit time [1]. In contrast, a prolonged LTT was not observed for ^99m^Tc-pertechnetate administered into the spleen during trans-splenic portal scintigraphy (TSPS) [2]. Pulmonary transit time was significantly shorter at PET/CT [3] or radionuclide angiocardiography [4] compared to RHLT measured at PRPS. This suggests that the colorectal absorption of ^99m^Tc-pertechnetate plays a key role in slowing radiotracer flow through the liver and lungs, as seen in PRPS. Technetium is an artificial compound and a synthetic transition metal with high density. Consequently, we hypothesize that the reduced hepatic and pulmonary transit rates of colorectal-absorbed ^99m^Tc-pertechnetate may result from a not-yet-described defensive response activated against unidentified substances absorbed from the gut, which the body perceives as potentially hazardous chemicals (PHCs). This physiological response likely targets PHCs, including heavy metals and similar substances, which are accidentally ingested in food and drink [5,6]. A broad spectrum of PHCs can be absorbed through the colorectal mucosa, including toxic metals, pesticides, microplastics, heterocyclic amines, and polycyclic aromatic hydrocarbons from cooked foods. Rapid elimination of these hazardous compounds from the circulating blood is vital for the body’s health.

Conversely, a sudden decrease in portal flow, similar to what is observed with ^99m^Tc-pertechnetate in PRPS, has been reported during the early stages of liver metastasis in colorectal cancer (CRC) [7], but not in extraintestinal cancers [8]. The activation of this early portal-to-venous vasoconstriction, likely specific to CRC metastasis, appears closely associated with soluble substances released from the colorectal wall during tumour progression [9], resembling the anti-PHC response triggered at PRPS against ^99m^Tc-pertechnetate. This phenomenon has been used to diagnose occult CRC metastases by measuring an increased hepatic perfusion index (HPI) and Doppler perfusion index (DPI). However, its underlying physiopathology remains unclear. The dependence on operator skill and the need for expert technicians to perform the imaging methods used to calculate HPI or DPI [10,11,12], along with a limited understanding of the mechanism behind the early decrease in portal flow during CRC metastasis, have restricted the use of this diagnostic approach in current practice.

Our study aimed to examine the first-pass vasoactive response in the liver and lungs during PRPS with RBCs in vivo labelled with colorectal-absorbed ^99m^Tc-pertechnetate. Using dynamic data from time–activity curves (TACs) generated during liver angioscintigraphy (LAS) with ^99m^Tc-HDP (hydroxyethylene-diphosphonate), we sought to determine whether a radiotracer reaching the gut mucosa via arterial flow and then the liver via portal flow can trigger the anti-PHC response in the liver. Additional dynamic investigations utilizing various radiotracers and administration routes, included in the research, were conducted to highlight features of the anti-PHC mechanism. We also analyzed the hypothesis that pathological activation of this defensive response may favour CRC metastasis by contributing to an early reduction in portal flow.

## 2. Materials and Methods

All patients and healthy volunteers provided an informed consent statement before participation, and the confidentiality of their data was maintained throughout the study. The hospital’s ethics committee approved the nuclear medicine procedures and the scientific use of the laboratory database (approval code 4/2020, dated 29 January 2020). The procedures involved subjects aged 18 to 85 who were required to fast for 12 h prior to participation.

The LAS studies included patients referred for thyroid or bone scans who received as a bolus the standard dose of radiotracer for their scan. PRPS included patients referred for clinical purposes. This selection enabled RHLT and LTT assessment without exposing other subjects to unnecessary radiation.

### 2.1. Methods

#### 2.1.1. PRPS with In Vivo ^99m^Tc-Labelled RBC

We conducted a specialized type of PRPS to compare LTT and RHLT measured for in vivo ^99m^Tc-labelled RBC with the results for ^99m^Tc-pertechnetate from a previous study [1]. This method is a variant of the PRPS technique proposed and improved by Shiomi S. et al. [13,14].

Twenty to thirty minutes after intravenously administering 0.03 mL/kg of stannous pyrophosphate, a solution containing 2 mL of ^99m^Tc-pertechnetate (296–370 MBq) was introduced into the upper rectum, followed by 15 mL of air under pressure. Serial images were recorded every 2 s for 3 to 5 min. Each patient underwent two enemas, one the previous evening and another two hours before the examination. We note that the automatic effects of rectal instrumentation cannot influence liver and lung transit times (TTs) at PRPS, as they do not change the biochemical properties of ^99m^Tc-pertechnetate and do not impact the colorectal wall. The patients were positioned supine, with the camera detector covering the liver and heart regions. TACs were constructed for the liver and heart. The evaluation methods for LTT and RHLT are shown in Figure 1 and Figure 2, illustrating the moments when the radiotracer arrives: at the liver via the portal vein (T_L_), at the right heart (T_RH_), and at the liver via the hepatic artery (T_HA_).

An example of a cirrhotic patient where both types of RHLT calculations were used is shown in Figure 3. The left liver had both portal and arterial inflow, and RHLT was calculated using the left lobe TAC as shown in Figure 1. The right liver lacked portal inflow, so we used the RHLT method shown in Figure 2. Both values were around 42 s. We note that in areas without portal perfusion or with portosystemic shunts, LTT cannot be measured.

The PRPS studies were conducted using a Siemens Orbiter gamma camera (Erlangen, Germany) equipped with a high-resolution, low-energy parallel collimator, along with a Macintosh computer running IconView 9 dedicated software.

#### 2.1.2. LAS Performed with ^99m^Tc-Pertechnetate and ^99m^Tc-HDP to Determine RHLT

LAS was performed to measure RHLT for ^99m^Tc-pertechnetate injected intravenously as a bolus and to compare it with the values observed at PRPS.

We also conducted LAS with ^99m^Tc-HDP to determine if the range of RHLT values changes when ^99m^Tc bound to a radiopharmaceutical is given intravenously as a bolus.

To perform LAS with ^99m^Tc-pertechnetate and ^99m^Tc-HDP, we followed the classic methodology proposed by Leveson and Sarper [15,16]. The radiotracer was injected as a bolus into an antecubital vein using a solution volume of less than 0.5 mL. Patients were positioned supine. After the intravenous bolus administration of the radiotracer into the antecubital vein, we recorded serial anteroposterior and posteroanterior images of the chest and upper abdomen every second for one minute. Regions of interest (ROIs) were drawn on the cardiac area, right liver, and one of the kidneys, at least three times in each case, to verify the accuracy of the ROIs’ delineation and the bolus injection. We did not use ROIs on the left liver because of its increased arterial supply, observed even in healthy subjects. The quality of the bolus was verified by measuring the time from the half-maximum value to the peak of the renal curve, which must be under 8 s [17]. RHLT was calculated at LAS for both types of radiotracers as the time difference between the onset of the right liver TAC and the previous onset of the cardiac curve.

We performed LAS studies using an AnyScanS gamma camera (Mediso, Budapest, Hungary) equipped with a high-resolution, low-energy parallel collimator, running Nucline v2.02 and InterView^TM^ XP dedicated software.

#### 2.1.3. LR-TMDS Performed with ^99m^Tc-Pertechnetate

To assess the role of ^99m^Tc-pertechnetate absorption through the colorectal mucosa at PRPS, we introduced lower rectum transmucosal dynamic scintigraphy (LR-TMDS). Using this PRPS variation, we measured the RHLT of ^99m^Tc-pertechnetate injected into the submucosa of the lower rectum, bypassing the mucosa. Before reaching the right atrium, the radiotracer sequentially passed through the inferior rectal veins, internal and common iliac veins, and inferior vena cava. This vascular pathway mirrored that seen in patients with complete portal vein occlusion, as examined in PRPS (Figure 4).

A dose of 150–160 MBq of ^99m^Tc-pertechnetate was slowly injected into the submucosa of the lower rectum, in a volume of 0.5 mL, using a proctoscope positioned 5 to 8 cm from the anus, two hours after an enema. We then captured serial anteroposterior and posteroanterior images of the chest and upper abdomen at two-second intervals for four minutes. RHLT was determined as the difference between the onset of the TAC for the left kidney and right heart regions (Figure 5). We focused on the start of the left kidney’s TAC, as the radiotracer reached the kidneys and liver simultaneously; however, the beginning of the hepatic curve was less distinct. We conducted the LR-TMDS study using the same AnyScanS gamma camera as for LAS.

#### 2.1.4. LAS Performed with ^99m^Tc-HDP to Determine LTT

To assess whether vasoconstriction slows the flow of ^99m^Tc-HDP through the liver after intravenous injection and its arrival in the portal circulation following passage through the gut mucosa, we measured LTT using TACs constructed at LAS. We used a technique similar to that for RHLT measurement at LAS, while capturing anteroposterior and posteroanterior images of the abdomen (Figure 6 and Figure 7).

The TACs generated at LAS helped us determine when the radiotracer delivered through portal inflow reached the right liver (matching the peak of the left or right kidney dynamic curve, TP_LK_ or TP_RK_) and when the portal inflow of radiotracer started leaving the liver (TP_L_, indicated by the peak of the dynamic curve constructed on the right liver). LTT was recorded as the time between these two moments. (Figure 6 and Figure 7). We used TACs based on the aorta to verify the accurate delivery of the radiotracer bolus. TACs constructed on the spleen were analyzed to exclude chronic liver disease.

### 2.2. Study Population

#### 2.2.1. PRPS with In Vivo ^99m^Tc-Labelled RBC

PRPS with in vivo ^99m^Tc RBC labelling was performed on six healthy volunteers and 17 patients diagnosed with liver haemangiomas identified by ultrasound or CT, who were referred for differential diagnosis using labelled RBC. One control case, which showed LTT = 33 s, was diagnosed with acute hepatitis A and was excluded; the remaining control group included three men and two women aged 33 to 60. Three patients with haemangiomas larger than 4 cm in diameter were excluded because of the potential presence of arteriovenous shunts. The remaining patients with haemangiomas included nine men and five women, aged 27 to 61. Additionally, 26 patients with liver cirrhosis, including five with portal vein occlusion, who were referred to assess the possibility of visualizing portosystemic shunts, were examined to measure RHLT. We excluded 11 cirrhotic patients without portal occlusion. In four of these, enemas were replaced experimentally with colorectal cleansing using an osmotic laxative containing polyethylene glycol, which showed low colorectal absorption and improper resolution of TACs. Seven others were excluded because the upward inflection of the liver curve—determined by the arrival of radiotracer through the hepatic artery—was not precise, and RHLT could not be determined. The remaining group of cirrhotic patients included 15 men and six women, aged 48 to 70.

#### 2.2.2. LAS Performed with ^99m^Tc-Pertechnetate and ^99m^Tc-HDP to Measure RHLT

To evaluate RHLT for ^99m^Tc-pertechnetate intravenously injected, five healthy volunteers (two men and three women, aged 35 to 56) and 13 patients referred for thyroid scans underwent LAS with a dose of 150 to 185 MBq. Three patients were excluded, two with hyperthyroidism and one with heart failure. The remaining patients included seven women and three men aged 29 to 64.

To evaluate RHLT for ^99m^Tc-HDP, we performed LAS with a dose of 300–370 MBq, injected intravenously as a bolus in six healthy volunteers (four men and two women, aged 35–62 years) and 62 patients referred for bone imaging scans. Three patients were excluded because the difference between the TTs assessed by two experienced observers was about 15%. Four cases were excluded due to cardiac failure or chronic pulmonary disease. Eleven patients were excluded because of high fluctuation of the dynamic curves or incorrect bolus administration, and another 12 could not be included because the upward inflection of the liver curve at the radiotracer’s arrival through the hepatic artery was not detectable. The final analysis included six controls and 32 patients: 18 men and 14 women, aged 44 to 83 years and 33 to 72 years, respectively.

#### 2.2.3. LR-TMDS Performed with ^99m^Tc-Pertechnetate

LR-TMDS was performed in four volunteers without any lower rectum pathology: one woman and three men, aged 41 to 58 years.

#### 2.2.4. LAS Performed with ^99m^Tc-HDP to Measure LTT

LAS dynamic curves were analyzed to determine LTT for ^99m^Tc-HDP after intravenous injection, when it reaches the portal circulation following passage through the mesenteric arteries and gut mucosa. We examined five healthy volunteers (three men and two women, ages 40 to 61 years) and 61 patients referred for a bone scan with ^99m^Tc-HDP. Twenty-one patients were excluded from the study for the following reasons. In 12 of them, the hepatic curve showed multiple peaks, preventing LTT measurement. Five patients had chronic liver disease, one patient had an intracardiac shunt, one patient showed a 5 cm diameter haemangioma, and two patients had previously undergone hepatic surgery. The remaining study group included five healthy subjects and forty patients (15 women, aged 33 to 70 years, and 25 men, aged 42 to 82 years).

## 3. Results

### 3.1. PRPS with In Vivo ^99m^Tc-Labelled RBC

LTT ranged from 23.5 to 25.5 s in both healthy volunteers and patients with hemangiomas. This range was similar to that in our previous PRPS study using ^99m^Tc-pertechnetate, where the normal LTT was between 23 and 25 s (Figure 8). This confirms the activation of hepatic vasoconstriction in response to ^99m^Tc-pertechnetate after colorectal absorption, including for ^99m^Tc-labelled RBC.

RHLT values ranging from 39.5 to 43.5 s were measured at PRPS using in vivo RBC labelling in healthy controls, patients with haemangiomas, and patients with cirrhosis without portal vein occlusion. RHLT showed minor variation, regardless of the presence of chronic liver disease. In the five cirrhotic patients with portal occlusion who underwent this investigation, RHLT ranged from 40.5 to 42 s. These values were within the same range as the 41 to 43 s RHLT measured at PRPS with ^99m^Tc-pertechnetate. These results confirm that a selective pulmonary vasoconstriction is triggered in response to ^99m^Tc-pertechnetate after colorectal absorption, even when labelling RBC.

### 3.2. LAS Performed with ^99m^Tc-Pertechnetate and ^99m^Tc-HDP to Measure RHLT

RHLT ranged from 9 to 13 s in LAS conducted with ^99m^Tc-pertechnetate and from 9.5 to 13.5 s in LAS with ^99m^Tc-HDP.

The range of RHLT values remained the same when we used LAS with ^99m^Tc bound to a radiopharmaceutical, compared to the values observed for ^99m^Tc-pertechnetate. The RHLT values measured at LAS were about three to four times lower than those in PRPS, indicating that ^99m^Tc-pertechnetate and ^99m^Tc-HDP injected intravenously as a bolus did not slow down during their initial pass through the lungs.

This indicates that ^99m^Tc-pertechnetate and ^99m^Tc-HDP do not cause first-pass pulmonary vasoconstriction when administered intravenously, suggesting that these radiotracers are not directly recognized as PHCs in the lungs.

### 3.3. LR-TMDS Performed with ^99m^Tc-Pertechnetate

RHLT measured in LR-TMDS (Figure 5) for ^99m^Tc-pertechnetate injected into the lower rectum submucosa ranged from 12 to 15 s. ^99m^Tc-pertechnetate reached the right atrium by passing through the inferior hemorrhoidal veins, iliac internal and common veins, and vena cava in sequence. This pathway was similar to that of ^99m^Tc-labelled RBC or ^99m^Tc-pertechnetate in PRPS performed in patients with complete portal vein occlusion, which resulted in a significantly longer RHLT, ranging from 39.5 to 43.5 s.

This finding indicates that absorption through the colorectal mucosa plays a key role during PRPS in the vasoconstriction triggered in response to ^99m^Tc-pertechnetate. It suggests that the vasoconstriction mediators, which directly cause pulmonary vasoconstriction at PRPS, are released from the mucosa. It also confirms the LAS finding that ^99m^Tc-pertechnetate is not directly recognized in the lungs as a PHC. Consequently, the vasoconstriction mediators released from the colorectal mucosa, rather than ^99m^Tc-pertechnetate, are the direct triggers of the selective vasoconstriction observed at PRPS in the lungs.

### 3.4. LAS Performed with ^99m^Tc-HDP to Measure LTT

LTT values measured at LAS for ^99m^Tc-HDP, which is delivered to the gut mucosa via arterial flow following intravenous injection and reaches the liver through portal inflow, ranged from 20 to 27 s, consistent with the values between 23 and 25.5 s recorded at PRPS for colorectal-absorbed ^99m^Tc-pertechnetate and ^99m^Tc-labelled RBC. This indicates that vasoconstriction mediators are released from the mucosa not only when a PHC is absorbed from the gut but also when a PHC arrives at the mucosa via arterial flow. We note that the LTT measured at LAS stayed within this range even for patients with evident liver metastases from various cancers who did not have a portosystemic shunt (Figure 7).

### 3.5. Statistical Analysis

We examined, by measuring TTs using dynamic investigations, a mechanism that reveals two states: either liver or lung anti-PHC vasoconstriction occurs in response to ^99m^Tc derivatives, or it does not.

For LTT, we identified two ranges. Values below 8 s indicated a lack of vasoconstriction in the liver, while values above 19 s showed activated vasoconstriction against the radiotracer.

For RHLT, we also established two ranges. Values over 38 s indicated pulmonary vasoconstriction activation, while those under 15 s signalled the absence of pulmonary vasoconstriction.

The significant differences between TT ranges indicating presence or absence of vasoconstriction show high statistical significance in both LTT and RHLT (*p* < 0.001). We did not observe RHLT values between these ranges during investigations with ^99m^Tc-based radiotracers. In healthy volunteers, LTT values below 19 s were not observed. Values between 8 and 16 s reported for LTT in other imaging studies using different agents may suggest less intense anti-PHC responses compared to those triggered by ^99m^Tc derivatives.

No correlation was found between the TTs we analyzed and sex or age.

## 4. Discussion

### 4.1. Activation of the First-Pass Vasoactive Response in the Liver and Lungs to Facilitate the Removal of PHCs Absorbed from the Gut

In PRPS using in vivo ^99m^Tc-labelled RBC or ^99m^Tc-pertechnetate, the values obtained for LTT in healthy individuals were significantly higher than those observed in TSPS [2], magnetic resonance imaging (MRI) [18], and contrast-enhanced ultrasound [19,20,21,22]. Similarly, RHLT values measured at PRPS, with in vivo ^99m^Tc-labelled RBC or ^99m^Tc-pertechnetate, were significantly higher than in LAS, LR-TMDS, and the pulmonary transit time measured during PET/CT [3], radionuclide angiocardiography [4], or MRI [23]. (see Table 1 and Table 2).

The LTT between 23 and 25.5 s of colorectal-absorbed in vivo ^99m^Tc-labelled RBC and ^99m^Tc-pertechnetate was approximately four times longer than the LTT of 5–6 s observed at TSPS by Gao L et al. for ^99m^Tc-phytate injected into the spleen [2].

The much shorter transit times (TTs) through the liver for ^99m^Tc-pertechnetate, which is not absorbed from the gut, as observed in TSPS, and through the lungs, noted in LAS, LR-TMDS, and radionuclide angiocardiography, indicate that ^99m^Tc-pertechnetate does not directly cause vasoconstriction in these organs. This suggests that other substances are the direct triggers of vasoconstriction during PRPS. We hypothesize that ^99m^Tc-pertechnetate binds to specific vasoactive agents (VAs) during absorption through the colorectal wall, forming ^99m^Tc-VA soluble complexes that then directly activate vasoconstriction in the liver and lungs. The significantly prolonged RHLT observed during PRPS suggests that the vasoactive properties of ^99m^Tc-VA complexes remain unchanged after passing through the liver.

W. Wayne Lautt stated that resistance to blood flow in the liver primarily occurs in the hepatic venules. Due to the low gradient, approximately 6–10 mmHg, between the pressure in the portal and hepatic veins, vasoconstriction, which is activated in the hepatic venules, results in a sharp decrease in portal flow [24]. VAs released from the gut mucosa might cause vasoconstriction in the hepatic venules, significantly reducing flow velocity through sinusoids and thus helping Kupffer cells eliminate PHCs. The arterial inflow could be slightly affected by the increased pressure in post-sinusoidal venules, which may be activated during the passage of PHC-VA complexes, due to the high mean pressure in the hepatic artery, similar to that in the aorta [25]. Other vasoactive mechanisms may also be activated in the hepatic sinusoids [26,27,28,29,30]. Some of them increase cell adhesion and may decrease portal flow velocity [31,32,33]. Similarly, we hypothesize that PHC-VA complexes cause vasoconstriction in pulmonary arterioles, significantly reducing their flow through the pulmonary capillaries, thus improving the removal of PHCs by the lungs’ mononuclear phagocyte system [34].

The vasoactive response may occur in the small vessels of the liver and lungs that are crossed by ^99m^Tc-VA complexes. In the liver, vasoconstriction can theoretically be triggered in many of the approximately 50,000 post-sinusoidal venules. The liver acts as a hemodynamic buffer, and a temporary reduction in hepatic blood flow can happen alongside a markedly decreased velocity of the radiotracer flow between the portal and hepatic veins. In contrast, the significant increase in RHLT to about 42 s observed in PRPS cannot be attributed to blood flow but only to radiotracer flow, as such a sudden and substantial decrease in pulmonary blood flow velocity would cause shock. We presume that ^99m^Tc-VA complexes pass through and induce vasoconstriction in a small fraction of the millions of pulmonary arterioles, leaving the overall blood flow velocity through the lungs unchanged.

^99m^Tc and its derivatives are safe radiotracers that are well tolerated after intravenous administration. However, ^99m^Tc-pertechnetate is a synthetic compound, and must be recognized by the body during colorectal absorption as a foreign substance, making it a PHC. Consequently, the vasoactive first-pass defensive response is activated against it in the liver and lungs.

An increase in RHLT is not observed in LR-TMDS, where ^99m^Tc-pertechnetate is not absorbed through the mucosa. This suggests that the VAs that bind to ^99m^Tc-pertechnetate, which we suppose directly activate hepatic and pulmonary anti-PHC vasoconstriction during PRPS, are released from the colorectal mucosa.

The body should activate this defensive response against foreign substances identified as PHCs throughout the digestive tract, where absorption occurs. The uptake of ^99m^Tc-pertechnetate in the gastric mucosa interferes with dynamic PRPS-like studies of the stomach; therefore, alternative imaging methods are necessary to detect the presence of the anti-PHC mechanism in the stomach.

Various oral medications and food components can theoretically activate the vasoconstriction mechanism in the liver and lungs if they are detected as PHCs during absorption from the gut, reducing their bioavailability by increasing elimination. Substances absorbed from the gastrointestinal tract that activate the anti-PHC response activate vasoconstriction and temporarily decrease blood flow velocity through the liver and selectively in the lungs. This weakens shear forces and may increase the risk of hepatic and possibly pulmonary metastases [31]. Consequently, patients with different types of cancer should avoid foods or medications that may contain substances the body could recognize as PHCs.

PRPS and TSPS conducted with ^99m^Tc-pertechnetate in dogs and cats showed a longer LTT in PRPS (average: 12 s in dogs and 14 s in cats) compared to TSPS (average: 7 s in dogs and 8 s in cats) [35,36,37]. This indicates that PHCs absorbed through the colorectal mucosa activate a first-pass vasoconstriction response in the livers of other mammals, not just in humans.

We note that in our study, we could not assess the intraindividual reproducibility of the data due to strict radiation protection guidelines that prevent repeating nuclear medicine investigations at short intervals without a specific clinical reason. However, the accuracy of data interpretation is supported by the low interindividual variability of TTs.

### 4.2. Activation of the Hepatic Vasoactive Response to PHCs Following Intravenous Administration, Reaching the Gut Mucosa via Arterial Flow, Demonstrates Its Overall Role in Removing PHCs from Circulating Blood

We performed a separate LAS analysis to determine whether the hepatic vasoconstriction is triggered in response to PHCs reaching the gut mucosa via arterial flow and then entering the liver through portal flow. This assessment aimed to find out if the activation of the anti-PHC mechanism occurs during the repeated recirculation of PHCs after they are absorbed from the gut, or if they enter the bloodstream through another pathway.

The LTT values ranged from 22 to 27 s for the radiotracer injected intravenously at LAS, consistent with the average of 24 s observed at PRPS. This supports the idea that the vasoactive response is triggered in the liver by PHCs from the portal flow, which previously reached the gut mucosa via arterial blood. Additionally, the similar range of LTT values at PRPS and LAS indicates a comparable strength of vasoactive responses caused by PHCs, whether absorbed from the gut or delivered to the gut mucosa through arterial blood. This suggests that the vasoactive mechanism we describe functions not only as a first-pass defensive response to PHCs absorbed from the gut but also as a broader process that helps remove substances recognized as PHCs from circulating blood. Additional research is needed to determine whether PHCs reaching the gut mucosa through arterial flow can trigger the anti-PHC response in the lungs.

Further research is required to determine whether specific imaging agents intravenously injected are recognized during their passage through the gut mucosa as PHCs, and vasoactive defensive responses of different strengths are triggered against them. This question arises from the notable differences in the mean portal vein to hepatic vein TTs seen in contrast-enhanced ultrasound after intravenous administration of various echo-enhancers: 6.33 s for BR1 [19], 6.45 s for Sonazoid [20], 15.27 s for Sonovue [21], and 9.597 s for Levovist [22]. The strength of the vasoactive responses and the LTT values likely depend on the chemical properties of the substances identified as PHCs in the intestinal mucosa.

### 4.3. Pathological Activation of the Anti-PHC Mechanism by VAs, Which Can Be Released from Primary Tumours and Disseminated Cells, Might Play a Role in Early CRC Metastasis

During the early development of CRC liver metastases, an increase in HPI measured at LAS, CT, or MRI, along with a similar rise in DPI calculated via ultrasound, is caused by a reduction in portal flow [38,39]. This phenomenon is attributed to circulating vasoactive agents [9,40,41]. Elevated HPI and DPI have shown high diagnostic value for detecting occult CRC liver metastases [15,42,43,44,45]. However, the underlying mechanism of this early reduction in portal flow linked to CRC metastasis remains unknown. Consequently, HPI and DPI are uncommonly used in diagnosing occult CRC liver metastases or related studies.

We observed that the reduction in portal flow during the early stages of CRC liver metastasis mirrors the decrease in portal flow velocity, accompanied by an increase in LTT, triggered by the anti-PHC mechanism, as seen in PRPS or LAS. Specific substances released from the colorectal mucosa appear to be crucial for both the anti-PHC vasoconstriction response highlighted at PRPS or LAS and the early metastasis of CRC. Over ninety percent of CRCs originate from the mucosal layer, which our study indicates typically secretes VAs involved in the anti-PHC response. The release of VAs from primary CRC tumours into portal tributaries may be related to impairment of the gut vascular barrier [46] and could precede the start of dissemination, facilitating it. VAs trigger the anti-PHC vasoactive response, reducing portal flow velocity, which weakens shear forces and thus promotes metastasis [31]. Besides vasoconstriction, narrowing of small vessels through other mechanisms may also promote early CRC metastasis by reducing flow velocity in the sinusoids and post-sinusoidal venules [47,48]. No increase in HPI was observed in occult liver metastases from cancers originating outside of the gut [8]. Our analysis suggests that extraintestinal cancers do not decrease portal flow during early liver metastasis because they lack cells that produce VAs and, therefore, usually do not activate the anti-PHC mechanism.

An early decrease in portal flow marks the first step in a series of hemodynamic changes linked to CRC metastasis. During the initial development of CRC liver metastases, portal flow likely drops due to a vasoactive response probably triggered by VAs, along with a moderate increase in hepatic arterial flow caused by the hepatic artery buffer response. As metastases advance, supported by neovascularization, portal flow remains low as primary tumours and metastatic cells continue releasing VAs. Meanwhile, hepatic arterial flow gradually increases and becomes the main factor driving the abnormal rise in HPI when CRC metastases exceed 2 mm in diameter [49]. This increased arterial flow causes significant rises in HPI or DPI in patients with clinically visible liver metastases not only from CRC but also from other cancers [50,51]. We suggest that CRC metastatic emboli, having detached from the colorectal mucosa and adhered to sinusoids or venules, can release VAs in situ, activating the anti-PHC mechanism. Consequently, even after surgical removal of the primary tumour, HPI and DPI levels may remain elevated in patients with occult CRC metastases [52].

Further research is needed to investigate potential activation of the anti-PHC response during the early metastasis of gastrointestinal cancers beyond CRC, considering the initial increase in DPI and HPI observed in gastric and oesophageal cancers [15,53,54]. Additionally, more studies are needed to identify the biochemical basis of the hemodynamic mechanisms we investigated.

### 4.4. Study Limitations

More research is needed to identify the VAs, which our analysis indicates can be obtained from portal blood flow after the per-rectal administration of ^99m^Tc-pertechnetate.

LR-TMDS could be performed with a small group of patients, since the procedure is uncomfortable for subjects and must be performed by a surgeon.

We were unable to perform upper rectum transmucosal dynamic scintigraphy using ^99m^Tc-pertechnetate administered Via endoscopy or conduct per-rectal portal scintigraphy with radiopharmaceuticals other than ^99m^Tc-pertechnetate.

Our study employed traditional nuclear medicine dynamic procedures to explore the liver, a method that many researchers and clinicians have discontinued, leading to a scarcity of recent articles on the subject.

Further studies are necessary to assess the potential of specific medications targeting VAs or their receptors, which may reduce liver and possibly lung CRC metastasis.

## 5. Conclusions

This study suggests that a first-pass vasoactive response is initiated in the liver and lungs during PRPS with ^99m^Tc-labelled RBC or ^99m^Tc-pertechnetate to help remove ^99m^Tc-pertechnetate, which is a synthetic compound that the body recognizes as foreign, and a PHC during colorectal absorption. We hypothesized that VAs released from the gut mucosa bind to PHCs, forming soluble complexes that directly activate a strong vasoactive response in the liver and lungs, significantly slowing PHC flow. LTT calculated at LAS with ^99m^Tc-HDP indicates that this defensive response is also triggered in the liver in response to unknown substances reaching the gut mucosa via arterial flow, aiding in the removal of PHCs from circulating blood.

A similar sudden decrease in portal flow occurs early in CRC liver metastasis. CRC cells mainly originate from the mucosa and typically produce VAs, which can be released into portal tributaries during tumour progression. We believe this process may contribute to CRC metastasis, as the VAs secreted by CRC mucosal cells cause vasoconstriction, reducing portal flow velocity and thereby aiding metastatic emboli in seeding and growth. Extraintestinal cancers generally do not produce VAs, so they do not trigger an anti-PHC response during their liver metastasis. As a result, DPI and HPI are ineffective for diagnosing occult liver metastasis from these cancers.

This study could influence areas within the pharmacokinetics, toxicology, and food additives industry, and may also benefit oncology, particularly in diagnosing and treating CRC. Monitoring VAs in the blood could aid in early detection and treatment of CRC with a high risk of metastasis.

## Figures and Tables

**Figure 1 diagnostics-15-02444-f001:**
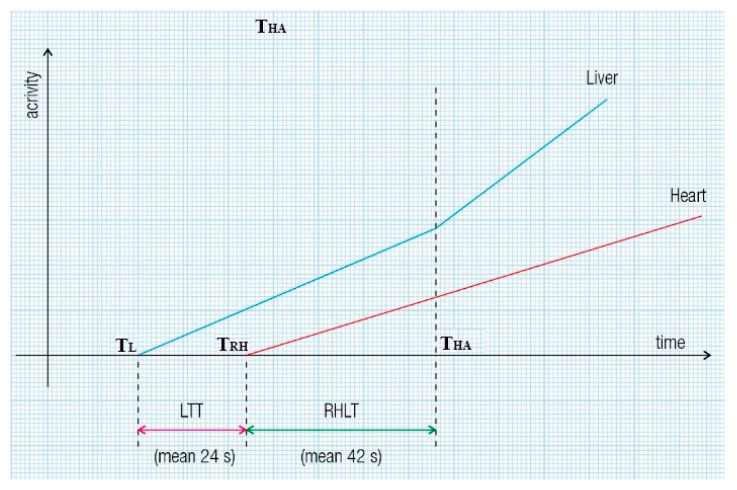
Liver transit time (LTT) and right heart to liver transit time (RHLT) assessment during per-rectal portal scintigraphy (PRPS) in healthy volunteers. LTT is calculated as T_RH_ minus T_L_, where T_RH_ indicates the time of radiotracer arrival in the right heart, and T_L_ indicates its arrival in the liver via the portal vein. To determine RHLT in liver regions with both portal and arterial inflow, we used the moment of the upward inflection of the liver’s time–activity curve (TAC) caused by the radiotracer reaching the liver through the hepatic artery (T_HA_): RHLT equals T_HA_ minus T_RH_.

**Figure 2 diagnostics-15-02444-f002:**
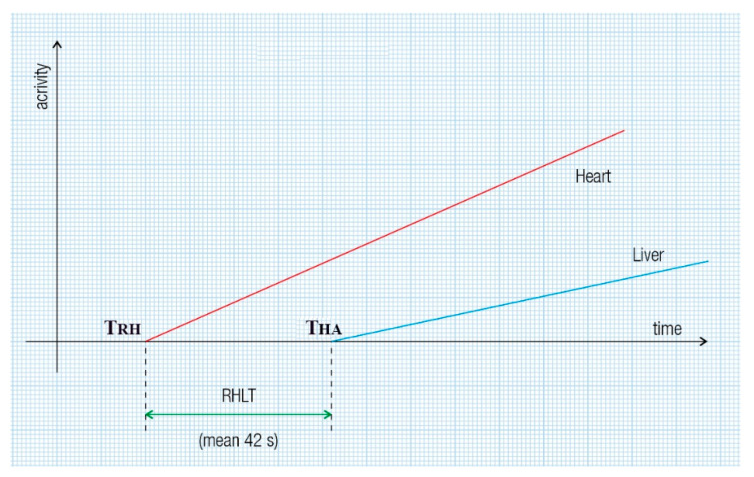
RHLT assessment at PRPS in patients with portal vein obstruction involves the radiotracer reaching the entire liver or a single lobe solely through the hepatic artery. RHLT is determined by subtracting the onset time of the right heart’s dynamic curve from that of the liver’s dynamic curve: RHLT = T_HA_ − T_RH_.

**Figure 3 diagnostics-15-02444-f003:**
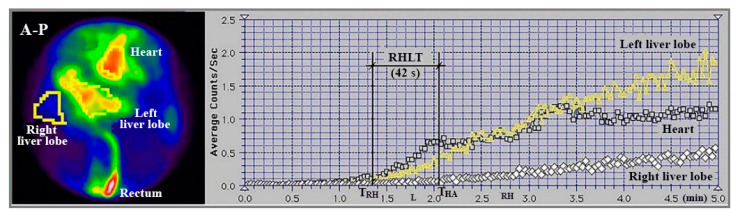
Summed image in antero-posterior view (A-P) and TACs generated during PRPS performed with ^99m^Tc-pertechnetate in a patient with alcoholic cirrhosis and complete occlusion of the right portal vein. RHLT = T_HA_ − T_RH_ = 42 s, as calculated for both lobes.

**Figure 4 diagnostics-15-02444-f004:**
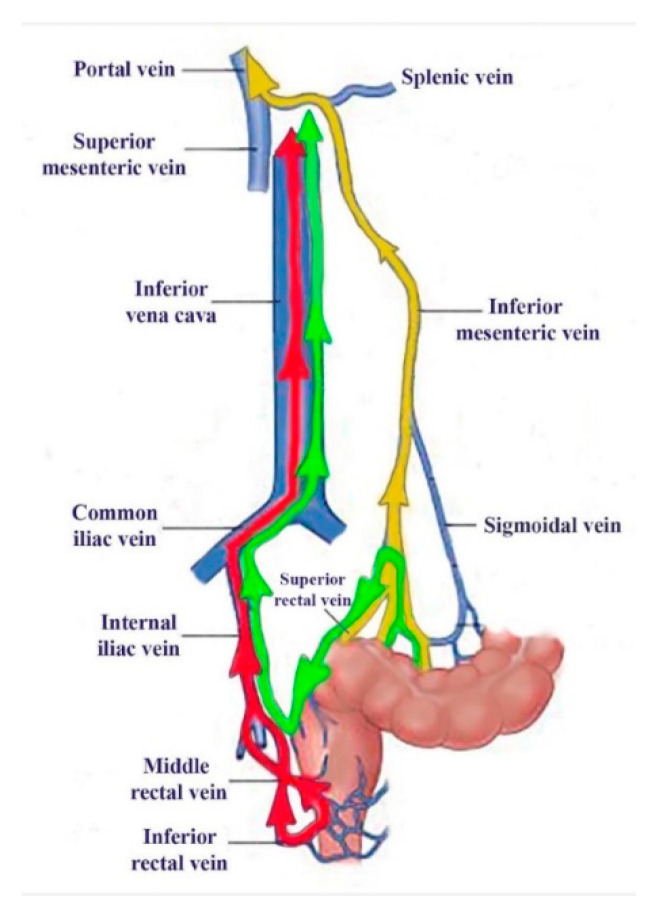
The vascular pathways of ^99m^Tc-pertechnetate during our investigations: colorectal absorbed in PRPS in healthy volunteers (yellow) and patients with complete portal vein obstruction (green), as well as after lower rectum transmucosal dynamic scintigraphy (LR-TMDS) injection into the submucosa of the lower rectum (red).

**Figure 5 diagnostics-15-02444-f005:**
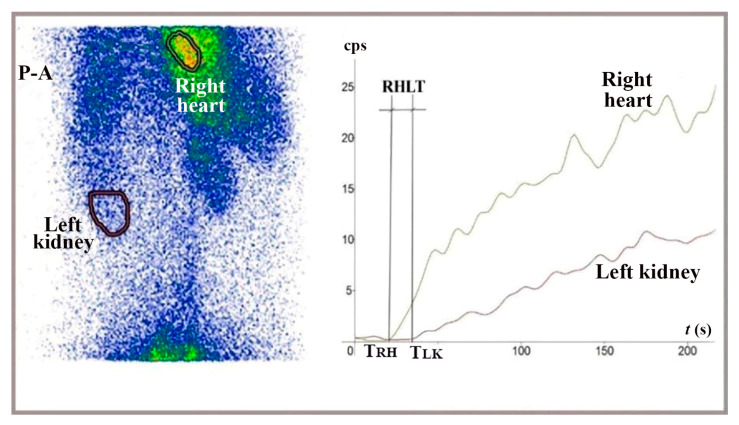
Summed image in posteroanterior view (P-A) and TACs generated for the right heart and left kidney during LR-TMDS, following the administration of ^99m^Tc-pertechnetate into the lower rectum submucosa. RHLT = T_LK_ − T_RH_ = 15 s.

**Figure 6 diagnostics-15-02444-f006:**
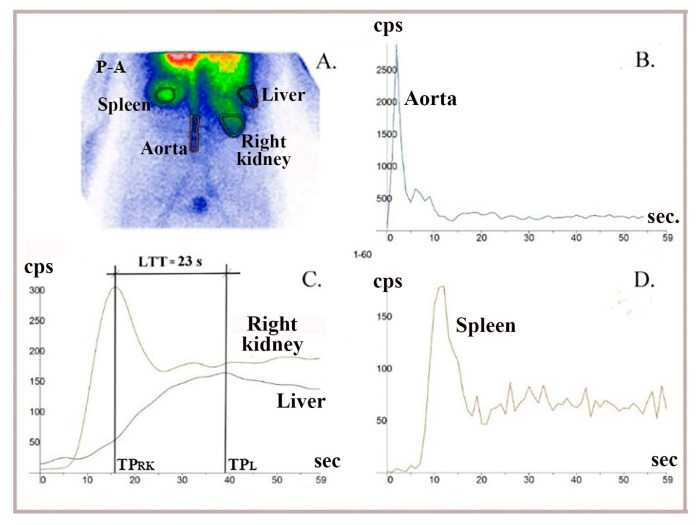
LAS performed using ^99m^Tc-HDP (hydroxyethylene-diphosphate) in a healthy patient. (**A**) Summed image in postero-anterior view. (**B**) A dynamic curve built on the aorta to control bolus quality. (**C**) The LTT = 23 s was determined as the difference between the peaks of the right liver lobe (TP_L_) and right kidney (TP_RK_) curves. (**D**) TACs built on the spleen were analyzed to exclude patients with possible changes in LTT due to chronic liver disease.

**Figure 7 diagnostics-15-02444-f007:**
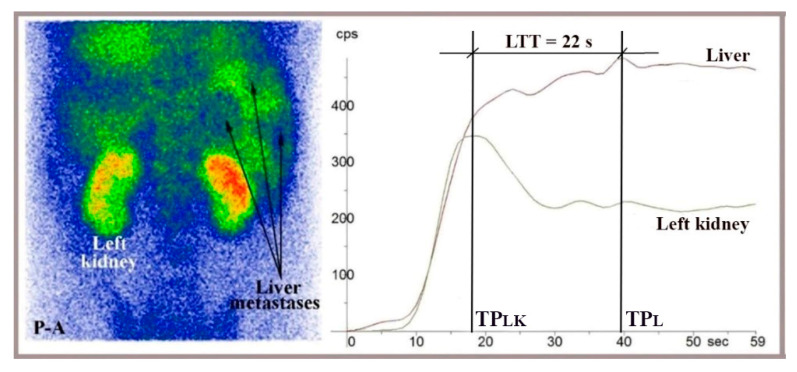
LAS performed in a posteroanterior view (P-A) with ^99m^Tc-HDP on a 44-year-old patient showing liver metastases from colorectal cancer, visible on the summed image. LTT was 22 s, calculated as the difference between the peaks of the right liver and left kidney TACs (LTT = TP_L_ − TP_LK_).

**Figure 8 diagnostics-15-02444-f008:**
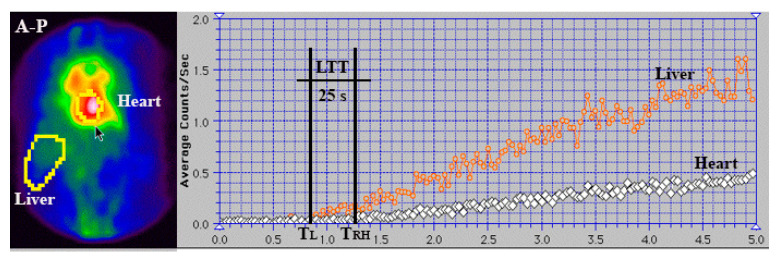
Summed image and TACs generated for the liver and heart regions during PRPS using ^99m^Tc-labelled RBC in a healthy volunteer. LTT was calculated as the difference between the starting points of the right heart (T_RH_) and liver (T_L_) curves. LTT = T_RH_ − T_L_ = 25 s.

**Table 1 diagnostics-15-02444-t001:** The significance of the results from the dynamic nuclear medicine procedures conducted in the study.

Sr. No.	The Dynamic Scintigraphy Method Performed in This Study	Transit Time Value Determined	Significance of the Results
1.	PRPS with RBC in vivo labelled with ^99m^Tc-pertechnetate.	Highly increased LTT (23.5–25.5 s) and RHLT (39.5–42.5 s)	First-pass vasoconstriction is triggered in the liver and lungs in response to colorectal-absorbed ^99m^Tc-pertechnetate, an artificial substance that is recognized during absorption as a potentially hazardous chemical (PHC).
2.	LAS conducted to evaluate RHLT for ^99m^Tc-pertechnetate intravenously injected as a bolus	Low value of RHLT (9–13 s)	^99m^Tc-pertechnetate and ^99m^Tc-HDP, when injected intravenously, are not recognized in the lungs during the first pass as PHCs and, as a result, do not directly induce pulmonary vasoconstriction.
3.	LAS conducted to evaluate RHLT for ^99m^Tc-HDP intravenously injected as a bolus	Low value of RHLT (9.5–13.5 s)	
4.	LR-TMDS conducted to evaluate RHLT for ^99m^Tc-pertechnetate administered in the lower rectum submucosa	Low value of RHLT (12–15 s)	Even if ^99m^Tc-pertechnetate crosses the other layers of the colorectal wall but is not absorbed through the mucosa, it does not directly activate first-pass vasoconstriction in the lungs. This indicates that the vasoactive agents responsible for vasoconstriction in PRPS are secreted in the colorectal mucosa.
5.	LAS conducted to evaluate LTT for intravenously injected ^99m^Tc-HDP, delivered to the gut mucosa through the arterial flow	Highly increased LTT (20–27 s)	Vasoconstriction also occurs in the liver when ^99m^Tc-HDP is delivered via arterial flow to the gut mucosa and then through portal flow to the liver.

**Table 2 diagnostics-15-02444-t002:** Comparative LTT, RHLT, and pulmonary transit time measured in healthy controls using various imaging methods.

Sr. No.	Imaging Method	LTT	RHLT	Pulmonary Transit Time
1.	PRPS with ^99m^Tc-pertechnetate [1]	23–25 s	41–43 s	-
2.	Trans-splenic portal scintigraphy (TSPS) with ^99m^Tc-phytate [2]	5–6 s	-	-
3.	PET/CT with [^82^Rb]RbCl [3]	-	-	5–13.4 s
4.	Pulmonary radionuclide angiography with ^99m^Tc-DTPA (diethylene-triamine-pentaacetate) [4]			6.2–9.2 s
5.	MRI [18]	3.03 s	-	-
6	MRI [23]	-	-	5.9–7.9 s
7.	Contrast-enhanced ultrasound utilizing different echo-enhancers [19,20,21,22]	6.33 s (BR1) 6.45 s (Sonazoid) 15.27 s (Sonovue) 9.597 s (Levovist)	-	-

## Data Availability

The raw data presented in this study are included in the article. Further inquiries can be directed to the corresponding author.

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
