# Peer review of "Hepatic and Pulmonary Vasoactive Response Triggered by Potentially Hazardous Chemicals After Passing Through the Gut Mucosa"

_diagnostics, 2025, doi:10.3390/diagnostics15192444_

Round 1

Reviewer 1 Report

Comments and Suggestions for Authors

This article hypothesizes that 99mTc-pertechnetate induces vasoactive substance release that leads to decreased portal flow when administered via the rectum, establishing a parallelism with what occurs in early CRC metastasis. The paper is interesting from a pathophysiological perspective, but there are several major flaws that need to be addressed (major changes), particularly regarding the lack of demonstration of causal mechanisms to support many of the affirmed conclusions. Moreover, the reference list is heavily outdated and require extensive changes to update the authors’ findings. Next, I detail some specific comments by sections.

Introduction:

In general, the introduction should be structured toward a verifiable hypothesis, namely the fact that 99mTC-pertechnetate induces portal flow vasoconstriction only when administered per-rectally. These findings could then be discussed with the parallelism of what occurs in CRC.

There is too much information concentrated into the first paragraph, which is excessively long. Such information should be divided into more paragraphs to facilitate reading and connection between ideas.

More elaborated definitions of LTT and RHLT (measurements and equations) should be provided.

Information regarding the mechanisms underlying the release of vasoactive substances in CRC should be further expanded, mentioning specific examples. On the other hand, the references used to support these findings (5-7) are very old (see specific comment on references below). More updated references are also needed so as to support the findings under current knowledge. Specifically regarding the “exclusivity” of this mechanism to CRC.

Methods and Results:

As written, cohorts and datasets are hard to understand and follow. There seem to be different groups (including data from a previous study). Please separate clearly what is new in this paper from what belongs to prior work (the latter should go in the introduction/discussion rather than the results). In this regard, I would recommend to add a simple flow diagram or a summary table showing, for each endpoint (e.g. RHLT, LTT): which cohort was used, inclusion/exclusion criteria, and how many participants were included and excluded (with reasons for further clarity).

The methods should include a short “Statistical analysis” subsection indicating which tests you used, how you summarized data (mean ± SD or median [IQR]), whether you report 95% confidence intervals (and maybe how you handled multiple comparisons and exclusions).

The Results section should be much more clearly presented. First, the first section (section 3.1, please see comment on sections numbering below) should be removed as apparently it only repeats previously published findings; present study’s results need to be separated from background material.

Second, for every analysis, report N at the start and N analyzed, with reasons for exclusion, and the main numbers with dispersion (mean ± SD or median [IQR]) and 95% CIs.

It would also be convenient to add a quick measure of reader agreement (even a simple metric helps).

Discussion

The observations and findings of the study do not allow to determine action mechanisms (e.g., vasoconstriction mediated by substance release), but only to suggest them, Thus, please soften causal assertions and language. The findings are related to variations in flow at different anatomical locations based on the administration route (and type) of radiotracers. This could be due to alternative mechanisms than those proposed by the authors, and biases should also be considered taking into account the low sample size (e.g., bolus dispersion or other hemodynamic modulations). Accordingly, the authors should focus on offering, at least with a short paragraph, alternative explanations (e.g., ROI choices, mixing of vascular compartments, autonomic effects of rectal instrumentation), and explain why you think the proposed mechanism remains plausible without claiming proof.

Similarly, caution should be exerted when linking the observed findings to early CRC metastasis. While I share that it as an interesting parallel, the paper does not demonstrate any specific pathway, thus further research is required.

References

Most references are really outdated. Only one paper from 2019 and another one from 2021 could be considered as relatively recent; the remaining 48 references date from >10 years old. This undermines the article, which needs to be discussed in the context of current knowledge. At least 80% references should be from the last 10 years, especially from the last 5 years.

Minor comments:

All subsections from section 3 (Results) and 4 (Discussion) are number as “1.1.”. Correct.

Review English and use of acronyms across the text. English is good in general.

Reviewer 2 Report

Comments and Suggestions for Authors

This investigation focuses on an understudied physiological defense system against potentially harmful substances (PHCs) ingested from the gut. It connects clinically significant processes, like early colorectal cancer (CRC) metastasis, to results from transit time measurements in nuclear medicine. In pharmacology, toxicology, and oncology, the idea that vasoactive agents (VAs) released from the intestinal mucosa mediate hepatic and pulmonary vasoconstriction is new and has the potential to be significant. 

Regarding the parts that can be modified, the hypothesis is highly speculative in parts, as the identity of the vasoactive agents remains unknown. Broader implications (pharmacokinetics, food additives, drug absorption) are discussed, but without supporting experimental evidence. Some clinical claims (e.g., prevention of metastasis by targeting VAs) remain premature given the current data. Small sample sizes in some subgroups (e.g., LR-TMDS performed on only four volunteers) limit statistical strength. Also, statistical analyses are limited (mostly descriptive, with occasional p-values). More rigorous tests (ANOVA, regression analysis) could strengthen conclusions. Some figures (e.g., Figures 3–6) would benefit from higher clarity and standardized legends. The overlap between the discussion and introduction sections makes the argument somewhat repetitive.

Round 2

Reviewer 1 Report

Comments and Suggestions for Authors

The authors have approached all my previous concerns and provided reasonable justifications to the main limitations of the article, including the need for citing relatively old references. 

The article is now clear and could be accepted for publication.

Reviewer 2 Report

Comments and Suggestions for Authors

The authors made the changes I initially mentioned.